# The Primary Complete Mitochondrial Genome of the Lappet Moth *Brahmophthalma hearseyi* (Lepidoptera: Brahmaeidae) and Related Phylogenetic Analysis

**DOI:** 10.3390/insects12110973

**Published:** 2021-10-28

**Authors:** Shan Yang, Shangren Gao, Shiyu Cai, Zhiwen Zou, Tianrong Xin, Bin Xia

**Affiliations:** School of Life Sciences, Nanchang University, Nanchang 330031, China; yangshan0794@126.com (S.Y.); banshouren0011@163.com (S.G.); caishiyu_xiaoyuer@126.com (S.C.); zouzhiwen@ncu.edu.cn (Z.Z.); xintianrong@ncu.edu.cn (T.X.)

**Keywords:** *Brahmophthalma hearseyi*, mitochondrial genome, Bombycoidea, Lasiocampoidea phylogenetic analysis

## Abstract

**Simple Summary:**

In this paper, the complete mitochondrial genome (mitogenome) of *B. hearseyi* was sequenced using long-PCR and primer-walking methods. The results indicated that the mitogenome is a typical circular molecule that is composed of 15,442 bp. Phylogenetic analysis showed that *B. hearseyi* is clustered into Brahmaeidae, and the phylogenetic relationships are (Brahmaeidae + Lasiocampidae) + (Bombycidae + (Sphingidae + Saturniidae)). This study provides the first mitogenomic resources for the Brahmaeidae.

**Abstract:**

*Background*: *Brahmophthalma hearseyi* (Lepidoptera: Brahmaeidae) is widely distributed across China. Its larvae damage the leaves of many plants such as those belonging to the Oleaceae family, causing significant economic losses and seriously affecting the survival and reproduction of *Cervus nippon*; however, genetic data for this species are scarce. *Methods*: The complete mitochondrial genome (mitogenome) of *B. hearseyi* was sequenced using long-PCR and primer-walking methods. Phylogenetic analysis that was based on 13 PCGs and two rRNAs was carried out using the neighbor-joining and Bayesian interference methods. *Results*: The mitogenome is a typical circular molecule that is made up of 15,442 bp, which includes 13 protein-coding genes (PCGs), 2 ribosomal RNA (rRNA) genes, 22 transfer RNA (tRNA) genes, and an A + T-rich region (456 bp). All of the PCGs, except for COX1 and COX2, start with ATN codons. COX2 and ND5 use the incomplete termination codon T, and 11 other PCGs use the typical stop codon TAA. All tRNA genes, except for trnS1 and trnS2, display a typical cloverleaf structure; trnS1 lacks the “DHU” arm, whereas trnS2 exhibits two mismatched base pairs in the anticodon stem. Phylogenetic analysis showed that *B. hearseyi* is clustered into Brahmaeidae, and the phylogenetic relationships are (Brahmaeidae + Lasiocampidae) + (Bombycidae + (Sphingidae + Saturniidae)). *Conclusions*: This study provides the first mitogenomic resources for the Brahmaeidae.

## 1. Introduction

*Brahmophthalma hearseyi* (Lepidoptera: Brahmaeidae) is distributed in China, Sikkim, India, Burma, Indonesia, and other countries. Its larvae damage various plants, specifically Oleaceae plants. Most of these plants have important economic value; they are important medicinal plants such as *Forsythia suspensa* and *Ligustrum lucidum*, perfume plants such as *Osmanthus fragrans* and *Jasminum sambac*, and oil plants such as *Olea europaea*. A larva can eat up 16 to 25 leaves per day [1,2,3]. As such, an outbreak of *B. hearseyi* can cause serious damage to these plants and contribute to large economic losses. Our survey found a large number of *B. hearseyi* in Jiangxi Taohongling National Nature Reserve [4], which cause serious damage to Oleaceae plants and seriously impact the survival and reproduction of *Cervus nippon* (which is already a critically endangered animal in China) [5]. Despite its potential economic impact, little is known about *B. hearseyi* [3]. Furthermore, it is morphologically and ecologically very similar to *Brahmaea wallichii*, but the adult size is slightly smaller [3]. Molecular information should provide more evidence for its identification.

The animal mitogenome is a double-stranded circular DNA molecule and is commonly 15–20 kb in size. It encodes 37 conserved genes, including 13 protein-coding genes (PCGs), 2 ribosomal RNA (rRNA) genes, 22 transfer RNA (tRNA) genes, and noncoding control regions that regulate its transcription and replication [6,7]. Furthermore, the mitogenome represents maternal inheritance, with a highly conserved phylogeny, non-recombination, a small size, stable structure, and easy purification [8,9]. Thus, mitogenomes are studied in several fields, such as systematic classification, molecular phylogeny, molecular evolution, population genetics, systematic geography, and molecular markers [8,10]. With the development of molecular methods in recent years, an increasing number of mitogenomes have been sequenced. To date, more than 200 Lepidoptera mitogenome sequences are available [11,12,13,14].

Lepidoptera is one of the largest insect orders worldwide with more than 165,000 species. There are 10 species classification methods, although some of the classification ideas or classification concepts that are based on traditional morphological characteristics suffer from much controversy, especially regarding the taxonomic status of Bombycoidea, Lasiocampoidea, and Sphingoidea [15,16]. Most scholars classify Lasiocampidae as Bombycoidea, while some scholars regard it as independent Lasiocampoidea. The classification features of Bombycoidea are mainly composed of missing or degraded morphology of the adults, and there are different classification systems that are proposed by Scoble [17], Heppner [18], and Wang [19]. The systematic classification system mainly takes the characteristics of the male external genitalia and the morphological differences of adult wing patterns as an important basis for classification [20,21,22]. Other classification methods are based on morphological or molecular characteristics [23]. At present, there is no molecular sequencing of the mitogenomes of *B. hearseyi* and other species of Brahmaeidae. It is important to study the mitogenome of *B. listenseyi* to clarify the classification method of the Bombycoidea superfamily. In this study, the mitogenome of *B. hearseyi* was analyzed, and the taxonomic status of Bombycoidae, Lasiocampoidea, and Sphingoidea was discussed based on their mitogenomes. The results of this study can provide an important theoretical basis for their identification.

## 2. Materials and Methods

### 2.1. Sample Collection

Adult *B. hearseyi* were collected from the Pengze Taohongling Sika Deer National Nature Reserve in Jiangxi, China. Fresh samples were preserved in 100% ethanol and stored in a −20 °C refrigerator for subsequent DNA extraction.

### 2.2. DNA Extraction

DNA from the muscle and connective tissue of adult individuals was obtained using the Qiagen DNeasy Tissue Kit [24].

### 2.3. PCR Amplification and Sequencing

Partial fragments of the *COX1*, *COX3*, *ND4*, *CYTB*, and *12S* genes were amplified using a lepidopteran universal primer [25,26]. The sequences were aligned with nucleotide databases in NCBI to confirm their identity. The primers of five long fragments (*COX1*–*COX2*, *COX3*–*ND4*, *ND4*–*CYTB*, *CYTB*–*12S*, and *12S*–*COX1*) were designed according to the sequences of five short fragments that were sequenced using Primer Premier 5.0 software [27]. The 10 primer pairs are listed in Appendix A.

Next, 2 × Mix Taq DNA polymerase was used to amplify the fragments that were less than 1 kb, whereas LA Taq DNA polymerase was used for fragments that were more than 1 kb in size. PCR cycling conditions were as follows: initial denaturation for 4 min at 94 °C, 35 cycles of denaturation for 10 s at 98 °C, annealing for 30 s at 50–55 °C, elongation for 1–4 min at 72 °C, and a final elongation step at 72 °C for 5 min. The PCR products were examined through electrophoresis on a 1.0% agarose gel and stained with ethidium bromide.

### 2.4. Sequence Assembly, Annotation, and Analysis

Long- and short-fragment sequences were assembled using SeqMan software [28]. Based on the original sequencing peak figure, the results were manually corrected, the space partition was removed, and the errors in the bases were corrected to ensure sequence accuracy.

The PCGs, rRNA genes, and tRNA genes were identified using the MITOS Web Server (mitos.bioinf.uni-leipzig.de/index.py, accessed on 8 August 2021) [29]. ORF software through the NCBI website (https://www.ncbi.nlm.nih.gov/orffinder/, accessed on 8 August 2021) was also used to verify the annotated protein-coding genes.

The sequence length, base content, codon usage, and amino-acid composition were statistically analyzed using Editseq and MEGA 5.0 [30]. The bias of nucleotide composition was measured as AT skewness (AT skewness = (A − T)/(A + T)) and GC skewness (GC skewness = (G − C)/(G + C)).

The mitogenome sequence data of *B. hearseyi* were deposited in the GenBank database under the accession number KU884326.

### 2.5. Phylogenetic Analysis

A total of 13 PCGs and 2 rRNA genes of 20 lepidopteran mitogenomes and the mitogenome of *Drosophila melanogaster* were downloaded from GenBank. *D. melanogaster* was used as an outgroup. Two combined datasets (PCG and rRNA datasets) are concatenated with 13 PCGs and 2 rRNA genes of 22 mitogenomes in Table 1. These PCGs and rRNA genes were aligned using MEGA 5.0 and Clustal × 1.83 [31]. The phylogenetic analysis was based on 13 PCGs and 13 PCGs + 2 rRNAs using neighbor-joining (NJ) [32] and Bayesian inference (BI) [33] methods (Table 1).

## 3. Results

### 3.1. Genome Composition and Base Structure

The complete mitogenome of *B. hearseyi* was 15,442 bp in length (GenBank accession number KU884326), which is longer than the average length of sequenced lepidopteran mitogenomes ((Figure 1). The mitogenome encoded 37 genes, including 13 PCGs (*COX1*–*3*, *COB*, *NAD1*–*6* and *4L*, and *ATP6* and *8*), 22 tRNA genes, and 2 rRNA genes. Among them, 14 genes (including 4 PCGs, namely, *ND1*, *ND4*, *ND4L*, and *ND5*), 8 tRNA genes (*trnF*, *trnH*, *trnP*, *trnL1*, *trnV*, *trnQ*, *trnC*, and *trnY*), and 2 rRNA genes were encoded by the L-strand. The 23 remaining genes were encoded by the H-strand. Both the location and the structure of PCGs, tRNA genes, and rRNA genes were considered conserved.

There were six gene overlaps (1–25 bp in size) and 19 intergenic spacers (1–49 bp in length) in the mitogenome (Appendix A). The fragment with the most overlap occurred between *trnL2* and *rrnL*, and there was a 7 bp overlapping fragment between *ATP8* and *ATP6*. The longest spacer was present between *trnN* and *trnS1* (Appendix A).

The nucleotide composition of the *B. hearseyi* mitogenome was as follows: T (40.67%), A (40.13%), C (11.72%), and G (7.47%). The A + T content, consistent with the characteristic of a strong A + T bias in insect mitogenomes, was 80.81% across the entire mitogenome, 79.27% in PCGs, 81.82% in tRNA genes, and 83.87% in rRNA genes (Appendix A). GC skewness was considerably higher than that of AT skewness across the whole mitogenome. The values of AT and GC skewness were −0.007 and −0.221, respectively. Moreover, the T base content was higher than that of A, and the C base content was higher than that of G.

### 3.2. PCGs

The A + T content in PCGs was 79.21%. which was significantly higher than that of G + C. In addition, the A + T content at the third codon position was the highest (93.33%) (Appendix A), which is similar to that in other insect mitogenomes [6,34,35]. GC skewness was lower than AT skewness in 13 PCGs, which was in contrast to that across the whole mitogenome. AT skewness was −0.160, and GC skewness was 0.025. In lepidopteran mitogenomes, the AT skewness values of PCGs are all negative, while the values of GC skewness are either negative or positive.

The relative synonymous codon usage was calculated using MEGA5.0. The results showed that amino acids with two synonymous codons showed a high usage frequency of A or U at the third codon position (Appendix A).

The codon usage analysis showed the most frequently used codon was UUA (L), which was followed by AUU (I), UUU (F), AUA (I), AAU (N), and UAU (Y). Moreover, the analysis of the amino-acid composition showed that the Leu amount (15.56%) was the highest, followed by Ile (11.99%), Phe (10.02%), and Ser (8.95%). The Cys amount was lowest, at only 0.78% (Appendix A).

### 3.3. rRNA and tRNA Genes

The two rRNA genes (*rrnL* and *rrnS*) of the *B. hearseyi* mitogenome were 1377 and 780 bp in length and were located between *trnL* and *trnV* and between *trnV* and the A + T-rich region, respectively. The base composition analysis showed that the A + T content of rRNA genes was 83.87%. The A + T bias was obvious, and the AT skewness and GC skewness were −0.019 and −0.397, respectively. The length, location, and base composition of the two rRNA genes were similar to those of other lepidopteran insects (Appendix A) [6,7,35,36,37,38].

The total length of the 22 tRNA genes of the *B. hearseyi* mitogenome was 1480 bp, of which the longest was *trnK* with 71 bp, and the shortest was *trnY* with only 63 bp. All of the tRNA genes, except for *trnS1* and *trnS2*, were folded into a typical cloverleaf structure; *trnS1* lacked the “DHU” arm, whereas *trnS2* exhibited two mismatched base pairs in the anticodon stem. However, *trnL1* (CUN) and *trnA* contained a U–U mismatch in the recipient stem. This is consistent with the results that were obtained in other lepidopteran insects. DHU and TΨC stems were 3–9 bp in length, the anticodon stems were 9 bp, and *trnL2* was 11 bp. The DHU arm of *trnS1* was simplified as a loop, which is identical to that of other lepidopteran insects (Appendix A) [7,39,40].

### 3.4. A + T-Rich Region

The A + T-rich region of *B. hearseyi* had a length of 455 bp between *rrnS* and *trnM* (Appendix A), with a high A + T content of 95.6%, while the G + C content was only 4.40%. The AT skewness for the control region was slightly negative (−0.085), indicating a higher occurrence of T compared with A nucleotides. The location, size, and structure of the A + T-rich region in mtDNA were not conserved.

The A + T-rich region of the *B. hearseyi* mitogenome featured an “ATAGA” motif that is similar to most Lepidopteran mitogenomes, an 18-nucleotide poly-T stretch following the “ATAGA” motif, and two “TA” short tandem repeats (STRs). There were two copies of 237 bp tandem repeats that were found in the A + T-rich region of *B. hearseyi* using the tandem repeats finder (Appendix A).

### 3.5. Phylogenetic Analysis

BI and NJ trees using the complete mitogenome of 25 species sequences were computed using the best-fit model of GTR + G + I [41].

In our study, the phylogenetic trees were constructed using two datasets (13 PCGs and two rRNAs using the NJ and BI methods (Figure 2). The results of the four trees were almost identical. The two trees that were constructed using the NJ method exhibited the lowest bootstrap values (44 and 14). These results indicate that Brahmaeidae was at one distinct branch, while Sphingidae and Brahmaeidae were separated from the Bombycoidea superfamily. The results of the four trees were similar when branches with low bootstrap values were adjusted. The phylogenetic relationships among these five families were (Brahmaeidae + Lasiocampidae) + (Bombycidae + (Sphingidae + Saturniidae)). Hence, Brahmaeidae and Lasiocampidae exhibited a close genetic relationship.

## 4. Discussion

The mitogenome composition and base structure of *B. hearseyi* revealed similar results to other Lepidoptera insects [9,40,42,43,44]. In the Lepidopteran mitogenome, the number, location, the length of gene overlap, and the intergenic regions affect changes in the size of the entire mitogenome, with certain differences between species. The longest gene overlap regions (61 bp) and the longest intergenic regions (222 bp) were found in the mitogenome of *Adoxophyes honmai* [45]. In most Lepidoptera, there is a 7 bp length overlap between *ATP8* and *ATP6*. Studies have shown that there is polycistronic transcription of the *ATP8*/*ATP6* gene [46]. These overlaps are transcribed together, probably because the mRNA of the *ATP8* gene is too short, and the translation efficiency is very low [5]. Instead, the complete start codon and stop codon were found on the complementary strand of the 314 bp intergenic sequence of *Triatoma dimidata*, which may encode an unknown gene.

The nucleotide composition of the *B. hearseyi* mitogenome was as follows: T (40.67%), A (40.13%), C (11.72%), and G (7.47%). The A + T content, consistent with the characteristic of a strong A + T bias in insect mitogenomes, was 80.81% across the entire mitogenome, 79.27% in PCGs, 81.82% in *tRNA* genes, and 83.87% in *rRNA* genes (Appendix A). GC skewness was considerably higher than AT skewness across the whole mitogenome. The values of AT and GC skewness were −0.007 and −0.221, respectively. Moreover, the T base content was higher than that of A, and the C base content was higher than that of G.

In addition, the A + T content at the third codon position was highest (93.33%) (Appendix A), which is similar to that in other insect mitogenomes [6,34,35]. Most PCGs were initiated with a typical ATN start codon: ATG for *ATP6*, *COX3*, *ND4*, *ND4L*, *CYTB*, and *ND1*, and ATT for *ND3*, *ND5*, and *ND2*. The four other PCGs, namely, *COX1*, *COX2*, *ATP8*, and *ND6*, used CGA, GTG, ATC, and ATA as the start codons, respectively [6,7,41]. A total of 11 of the PCGs were terminated with the stop codon TAA, but *COX2* and *ND5* were terminated with the incomplete stop codon T. This phenomenon exists in all sequenced lepidopteran mitogenomes. The formation of an incomplete stop codon may be attributed to post-transcriptional modification, such as polyadenylation, during the mRNA maturation process [6,44,47].

The analysis of amino-acid composition showed that the Leu amount (15.56%) was the highest, followed by Ile (11.99%), Phe (10.02%), and Ser (8.95%). The Cys amount was lowest, at only 0.78% (Appendix A). Leu is a hydrophobic amino acid, which may be related to most proteins that are encoded by mitogenomes being transmembrane proteins. Among the 20 amino acids of Lepidoptera insects, Cys content is lowest, which indicates that the amino-acid usage bias of the mitochondrial protein genes of Lepidoptera insects is relatively severe. The length, location, and base composition of the two rRNA genes were similar to those of other lepidopteran insects [6,7,35,36,37,38,48] (Appendix A). DHU and TΨC stems were 3–9 bp in length, anticodon stems were 9 bp, and *trnL2* was 11 bp. The DHU arm of trnS1 was simplified as a loop, which was identical to that of other lepidopteran insects [7,40,41] (Appendix A).

The mitogenome A + T-rich region of most lepidopteran insects is located between *rrnS* and *trnM*, a variable location that is caused by frequent tRNA rearrangements [4]. The mitogenome A + T-rich regions of Lepidoptera insects show abundant content and length polymorphism, with an average A + T content of up to 93.5% [49]. This indicates that the polymorphism of the size of the insect mitochondrial A + T-rich region may be due to the insertion/deletion of short tandem repeats (<500 bp) or the size of the intergenic region [4,36]. The A + T-rich regions of *Somena scintillans* and *Triuncina daii* also revealed two “TA” short repeats [50,51]. The A + T-rich region tandem repeats may be caused by nonhomologous recombination that is caused by sliding replication or unequal exchange [52,53]. However, the typical poly-A structure was deleted near the end of *trnM*. As the A + T-rich region is a noncoding region, the degree of variation is high. These conserved structural units differ greatly in the insect mitogenome, which may be related to mtDNA replication and transcription [54].

In our study, the phylogenetic relationships among these five families were (Brahmaeidae + Lasiocampidae) + (Bombycidae + (Sphingidae + Saturniidae)) (Figure 2). Hence, Brahmaeidae and Lasiocampidae exhibited a close genetic relationship. Heppner and many taxonomists used traditional classification methods according to the morphology and structure and indicated Sphingidae as a superfamily [16]. Furthermore, Tillyard [55], Kuznetzov and Stekolnikov [56], and Minet [57] believed that Lasiocampidae independently belongs to the Lasiocampoidea superfamily. Our study supports that Brahmaeidae belongs to the Lasiocampoidea superfamily, instead of the Bombycoidea superfamily. Therefore, this study supports the classification that was proposed by Scoble [15] and Brock [58], whereby both Brahmaeidae and Lasiocampidae belong to the Bombycoidea superfamily.

## 5. Conclusions

The complete mitochondrial genome (mitogenome) of *B. hearseyi* was sequenced using long-PCR and primer-walking methods. Its biological taxonomic status was determined by phylogenetic analysis. This mitogenome is a typical circular molecule that is made up of 15,442 bp, which includes 13 protein-coding genes (PCGs), 2 ribosomal RNA (rRNA) genes, 22 transfer RNA (tRNA) genes, and an A + T-rich region (456 bp). All of the tRNA genes, except for *trnS1* and *trnS2*, display a typical cloverleaf structure; *trnS1* lacks the “DHU” arm, whereas *trnS2* exhibits two mismatched base pairs in the anticodon stem. In the A + T-rich region of the mitogenome, the “ATAGA” motif, two copies of 237 bp tandem repeats, and two “TA” short tandem repeats were found. Phylogenetic analyses showed that *B. hearseyi* is clustered into Brahmaeidae. The phylogenetic relationships are (Brahmaeidae + Lasiocampidae) + (Bombycidae + (Sphingidae + Saturniidae)). Therefore, this study enriches the molecular biology data of Brahmaeidae, providing a theoretical basis for the identification of Brahmaeidae, as well as for the control of *B. hearseyi*.

## Figures and Tables

**Figure 1 insects-12-00973-f001:**
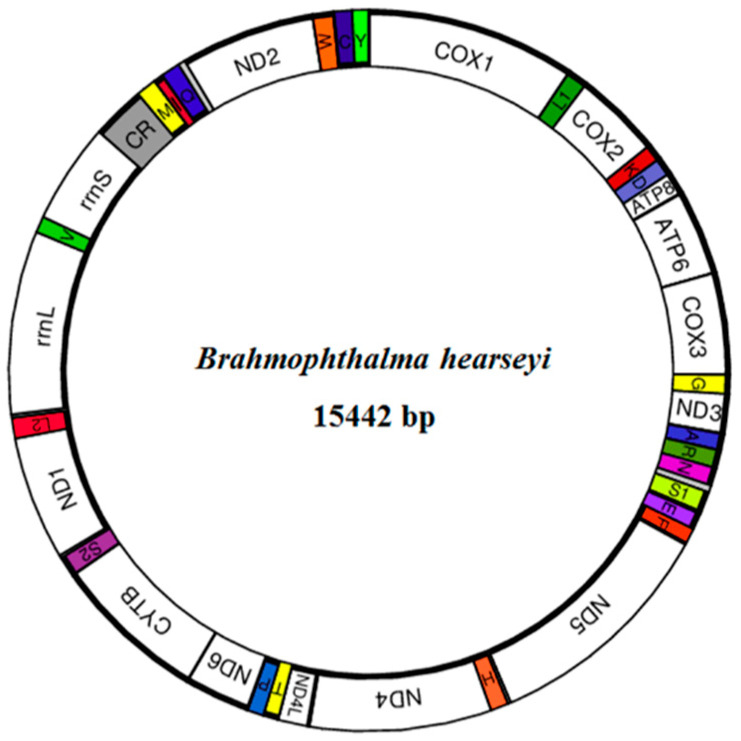
Circular map of the *Brahmophthalma hearseyi* mitochondrial genome. CR refers to the A + T-rich region.

**Figure 2 insects-12-00973-f002:**
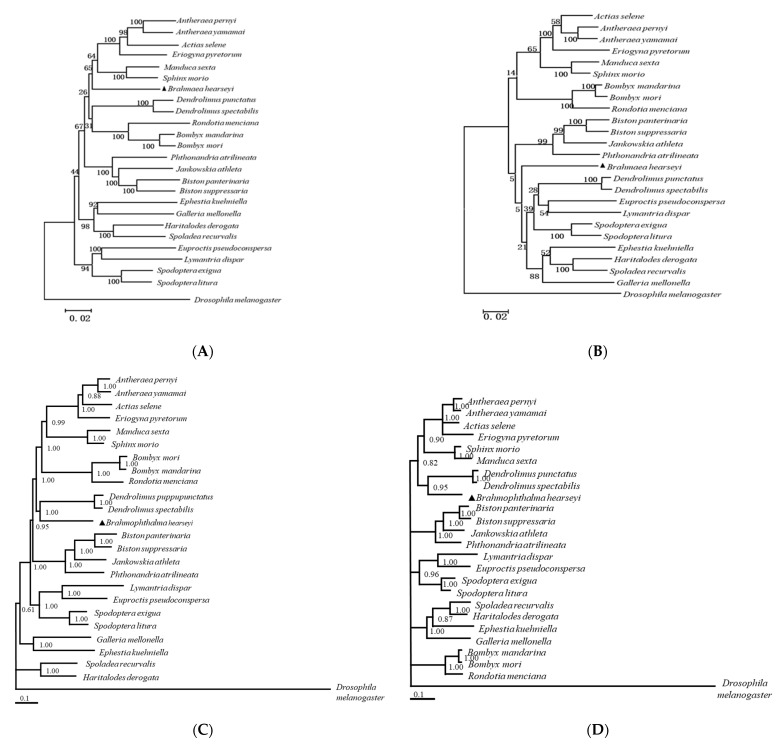
(**A**) Neighbor-joining tree based on the PCG dataset; (**B**) Neighbor-joining tree based on the rRNA dataset; (**C**) Bayesian inference tree based on the PCG dataset; (**D**) Bayesian inference tree based on the rRNA dataset.

**Table 1 insects-12-00973-t001:** The available insect mitogenomes (1 new and 24 obtained from GenBank) and their characteristics.

Family	Species	Whole Genome		PCG	rRNA	GenBank Accession No.	References
		Size	A + T	A + T	A + T		
		(bp)	(%)	(%)	(%)		
Saturniidae	*Actias selene*	15,236	78.91	77.36	83.73	NC_018133	Liu 2012
Saturniidae	*Antheraea pernyi*	15,566	80.16	78.47	83.87	NC_004622	Liu 2008
Saturniidae	*Antheraea yamamai*	15,338	80.29	78.94	84.14	EU726630	Kim 2009
Geometridae	*Biston panterinaria*	15,517	79.55	77.39	85.27	NC_020004	Yang 2012
Geometridae	*Biston suppressaria*	15,628	79.43	77.28	84.92	NC_027111	Chen 2015
Bombycidae	*Bombyx mandarina*	15,928	81.68	79.69	85.19	NC_003395	Yukuhiro 2002
Bombycidae	*Bombyx mori*	15,666	81.35	79.57	84.82	KM875545	Zhang 2014
Brahmaeidae	*Brahmophthalma hearseyi*	15,442	80.81	79.27	83.87	KU884326	This study
Lasiocampidae	*Dendrolimus punctatus*	15,411	79.46	77.62	84.73	NC_027156	Qin 2015
Lasiocampidae	*Dendrolimus spectabilis*	15,411	79.5	77.71	83.68	NC_025763	Tang 2011
Pyralidae	*Ephestia kuehniella*	15,295	79.76	78.18	84.44	NC_022476	Traut 2013
Saturniidae	*Eriogyna pyretorum*	15,327	80.82	79.41	84.55	NC_012727	Jiang 2009
Lymantriidae	*Euproctis pseudoconspersa*	15,461	79.93	77.99	84.87	NC_027145	Dong 2016
Pyralidae	*Galleria mellonella*	15,320	80.42	78.88	84.37	NC_028532	unpublished
Crambidae	*Haritalodes derogata*	15,235	80.7	79.19	84.59	NC_029202	Zhao 2015
Geometridae	*Jankowskia athleta*	15,534	79.53	77.71	83.83	NC_027948	Xu 2015
Lymantriidae	*Lymantria dispar*	15,569	79.88	77.84	84.6	NC_012893	Zhu 2010
Sphingidae	*Manduca sexta*	15,516	81.79	80.3	85.42	NC_010266	Cameron 2008
Geometridae	*Phthonandria atrilineata*	15,499	81.02	79.1	85.93	NC_010522	Yang 2009
Bombycidae	*Rondotia menciana*	15,301	78.86	77.1	83.74	NC_021962	Kong 2015
Sphingidae	*Sphinx morio*	15,299	81.17	79.84	84.8	NC_020780	Min 2013
Noctuidae	*Spodoptera exigua*	15,365	80.93	79.47	85.15	NC_019622	Qiu-Ling 2013
Noctuidae	*Spodoptera litura*	15,388	80.98	79.55	84.71	NC_022676	Wan 2013
Crambidae	*Spoladea recurvalis*	15,273	80.89	79.37	85.54	NC_027443	He 2014
Drosophilidae	*Drosophila melanogaster*	19,517	82.16	77.23	81.9	U37541	Clary 1982

## Data Availability

The data presented in this study are available in Appendix A.

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
