# Peer review of "The Primary Complete Mitochondrial Genome of the Lappet Moth Brahmophthalma hearseyi (Lepidoptera: Brahmaeidae) and Related Phylogenetic Analysis"

_insects, 2021, doi:10.3390/insects12110973_

Round 1
Reviewer 1 Report
Overall comments:
- The DNA sequence of the mitochondrial genome of B. hearsey provides valuable information. The phylogenetic data description and taxonomic discussion, however, are rather confusing and need to be improved.
- Extensive editing of English language is required.
Specific comments
- Brahmaeidae is called a subfamily (lines 36 and 319); Bombycoidea etc. (236-237) are called families, which is not according to taxonomic conventions; otherwise, the family names are misspelled.
- 18 and 38: Brahmaeidea is either a typo or an invalid butterfly superfamily
- Lasiocampidae is misspelled in all figs 5 to 8 (i is missing)
- 98-104: the description of the butterfly in Material and Methods is only relevant in case of possible confusion during collection. In that case, it sufficient to describe the properties used for discrimination of resembling species.
- Table 1: conditions referred to in the title are missing from the table. the meaning of “Size(kp)” is confusing (= size to the amplified fragment in kbp)?
- The DNA sequencing method used is missing in M&M (Sanger sequencing of PCR products? Equipment? Walking in case of large PCR fragments? outsourced?)
- 142: add unabbreviated terms for BI and NJ to this line instead of in line 149.
- 190: add abbreviation relative synonymous codon usage (RSCU) or add meaning of RSCU to the legend of Table 5.
- Fig 2: add tool + reference used to predict the RNA secondary structure
- 239 The results show in figure 5 (clustering of Brahmaeidae with Sphingidae) are not consistent with those of figs 6 to 8 (clustering with Lasiocampidae); yet the results in the 4 figs are said to be "nearly identical". Arguments for the final conclusion drawn on the family classification and the mixed use of family and superfamily names are confusing.
- 264-265: grammatical error
- 265-267: sentence to be reformulated
- 267-269: “Instead, the complete start codon and stop codon were found on the complementary strand of the 314bp intergenic sequence of Triatoma dimidata, which may encode an unknown gene”: it is not clear how this statement connects to the previous statements.
- 269-271: the concluding sentence of this paragraph is enigmatic.
- In the attached file, textual issues as well as confounding statements are highlighted in yellow. Not all issues are highlighted, however, since more extensive editing of English language is needed.

Author Response
We have made appropriate edits, edits and checks on the English language and style of the text, and made special revisions to the highlighted the parts of the manuscript that require special attention.Thank you very much for your suggestions. We have made changes to related issues. Please review them again. If there are other issues that need to be corrected (please refer to the attachment), I hope you can give us help and guidance. Thanks again.

Reviewer 2 Report
This manuscript consists of a mitogenome report for the moth Brahmaea hearseyi. The methods are well written and properly described in my opinion. The results are also well written, although I feel that at times the text was too descriptive, with repetition of information already present in tables and without any new interpretation of the results or clarification on why those features were important. The introduction and discussion would benefit from a substantial language revision.
The authors describe the mitogenome features extensively and perform phylogenetic analysis with protein coding genes as well as ribosomal RNA genes. The authors conclude that Brahmaea hearseyi clusters into the subfamily Brahmaeidae, although no other Brahmaeidae mitogenome was available and included in this analysis. In the absence of Brahmaeidae ingroups, I do not think this conclusion is meaningful or warranted.
Given the descriptive nature of this manuscript, I would recommend it it shortened, reviewed for language, and resubmitted as a Brief Report.
Author Response
Since the manuscript is mostly descriptive, it may be much shorter. In addition to reducing repetition, We reduce repetition and put Figures 2, 3, 4 and Tables 1, 3, 4, 5, and 6 in the supplementary materials.Thank you very much for your suggestion. We made changes to related issues. Please check them again. If there are other problems that need to be corrected (please refer to the attachment), I hope you can give us help and guidance. Thanks again.

Round 2
Reviewer 2 Report
In my opinion the revised manuscript has not sufficiently addressed the shortcomings of the original submission and has actually introduced new issues.
The abstract and introduction still need moderate English language revision. The phrase "and also the theoretical basis for the biological control of B.hearseyi." at the end of the Abstract (lines 22 and 43) might be an overstatement of the contributions of the manuscript.
The phylogenetic trees were built with PCGs and rRNAs. The tree branches should be rotated so that the similarities and differences in the layouts are more easily analyzed by the reader.
The authors have moved Table 1 to the supplement, but this table is still referenced in the main text as Table 1 when it should be "Supplementary Table 1". The main text starts with Table 2, which should have been renamed to Table 1 in the revised manuscript.
The authors have also moved Tables 3-6 to the supplement, but are still referencing these Tables in the main text without indicating that they are present in the supplementary file.
The supplementary materials include tables 1 and 3-6, but table 2 is missing. Likewise, supplementary figures 2 and 3 are included, but 1 seems to be missing.
I would recommend the authors carefully revise the manuscript presentation and text before submitting it, to make sure no such table/figure numbering errors are included.
Author Response
Comments 1 :The abstract and introduction still need moderate English language revision. The phrase "and also the theoretical basis for the biological control of B.hearseyi." at the end of the Abstract (lines 22 and 43) might be an overstatement of the contributions of the manuscript.
reply: The abstract and introduction We have undergone repeated English language and grammar revisions through English editing services and English-speaking colleagues. Although our original intention was that this study would provide a theoretical basis for the biological control of Brahmophthalma hearseyi, we fully accept your opinion and delete this part of the sentence.
Comments 2: The phylogenetic trees were built with PCGs and rRNAs. The tree branches should be rotated so that the similarities and differences in the layouts are more easily analyzed by the reader.
reply: We have rotated the branches of B in Figure 2 so that readers can more easily analyze the similarities and differences in the layouts.
Comments 3: The authors have moved Table 1 to the supplement, but this table is still referenced in the main text as Table 1 when it should be "Supplementary Table 1". The main text starts with Table 2, which should have been renamed to Table 1 in the revised manuscript.
reply: We carefully revised and checked the presentation and text of the manuscript to ensure that the table/figure numbers meet the requirements.

Round 3
Reviewer 2 Report
Please change B. hearseyi to italics in lines 141 and 157. Also, insert a space between the genus initial and the species name in B.hearseyi in line 309.
I would recommend another quick revision with attention to details such as the ones mentioned above. But I will no longer require to see the manuscript before it is published.
